# Risk of Hepatocellular Carcinoma after HCV Clearance by Direct-Acting Antivirals Treatment Predictive Factors and Role of Epigenetics

**DOI:** 10.3390/cancers12061351

**Published:** 2020-05-26

**Authors:** Luca Rinaldi, Riccardo Nevola, Gianluigi Franci, Alessandro Perrella, Giusy Corvino, Aldo Marrone, Massimiliano Berretta, Maria Vittoria Morone, Marilena Galdiero, Mauro Giordano, Luigi Elio Adinolfi, Ferdinando Carlo Sasso

**Affiliations:** 1Department of Advanced Medical and Surgical Sciences, University of Campania “L. Vanvitelli”, 80100 Naples, Italy; riccardo.nevola@unicampania.it (R.N.); aldo.marrone@unicampania.it (A.M.); mauro.giordano@unicampania.it (M.G.); luigielio.adinolfi@unicampania.it (L.E.A.); ferdinandocarlo.sasso@unicampania.it (F.C.S.); 2Department of Medicine, Surgery, Dentistry, University of Salerno “Scuola Medica Salernitana”, 84100 Salerno, Italy; gfranci@unisa.it; 3Immunological and Neurological Infectious Diseases, Cotugno Hospital, 80100 Naples, Italy; alex.perrella@hotmail.com; 4Department of Experimental Medicine, University of Campania “L. Vanvitelli”, 80100 Naples, Italy; giusycorvino1986@gmail.com (G.C.); mariavittoriamorone@gmail.com (M.V.M.); marilena.galdiero@unicampania.it (M.G.); 5Department of Medical Oncology, Centro di Riferimento Oncologico, Istituto Nazionale Tumori, 33081 Aviano, Italy; berrettama@gmail.com

**Keywords:** hepatocellular carcinoma, direct acting antivirals, HCV, cytokines, sustained virological response, epigenetic modulation

## Abstract

Direct-acting antivirals (DAAs) induce a rapid virologic response (SVR) in up to 99% of chronic hepatitis C patients. The role of SVR by DAAs on the incidence or recurrence of hepatocellular carcinoma (HCC) is still a matter of debate, although it is known that SVR does not eliminate the risk of HCC. In this review, we made an updated analysis of the literature data on the impact of SVR by DAAs on the risk of HCC as well as an assessment of risk factors and the role of epigenetics. Data showed that SVR has no impact on the occurrence of HCC in the short–medium term but reduces the risk of HCC in the medium–long term. A direct role of DAAs in the development of HCC has not been demonstrated, while the hypothesis of a reduction in immune surveillance in response to the rapid clearance of HCV and changes in the cytokine pattern influencing early carcinogenesis remains to be further elucidated. HCV induces epigenetic alterations such as modifications of the histone tail and DNA methylation, which are risk factors for HCC, and such changes are maintained after HCV clearance. Future epigenetic studies could lead to identify useful biomarkers and therapeutic targets. Cirrhosis has been identified as a risk factor for HCC, particularly if associated with high liver stiffness and α-fetoprotein values, diabetes and the male sex. Currently, considering the high number and health cost to follow subjects’ post-HCV clearance by DAAs, it is mandatory to identify those at high risk of HCC to optimize management.

## 1. Introduction

Treatment based on direct-acting antivirals (DAAs) has radically changed the natural history of chronic hepatitis C virus (HCV) infection [1]. In fact, while the previous therapeutic regimens based on the use of Interferon (IFN) were characterized by a sustained virological response (SVR) of 40–50%, DAAs allow an SVR almost in 100% of patients [2,3,4]. The result is of utmost clinical importance as HCV clearance is expected to prevent most of the serious complications due to progression of chronic hepatic C, as well as to the associated immune system dysregulation and chronic systemic inflammatory response [5,6,7,8,9,10,11]. However, DAAs are likely to affect the immune response, but the underlying biological mechanisms remain largely unexplored [12,13].

Hepatocellular carcinoma (HCC) is an important complication of HCV-related cirrhosis and it is reported with an average incidence of 1–4% per year [14]. The appearance of HCC is mainly related to two conditions: the first one is cirrhosis per se, with its necro-inflammatory activity, and the second one is the failure of immune surveillance with escape mechanisms [15,16,17,18].

In addition, HCV proteins exert a direct carcinogenic effect by deregulating the host cell cycle checkpoints and increasing the immune-mediated oxidative stress, which in turn leads to an increased DNA mutations frequency in the liver cells [19]. Thus, HCV clearance by DAAs and the consequent reduction in the hepatic necro-inflammation activity should reduce the risk of developing HCC.

Several long-term studies in HCV patients treated with IFN-based regimens have documented a reduction in the incidence of HCC by 75% in patients with SVR and a residual risk of the development of HCC mainly associated with some comorbidities such as metabolic syndrome and diabetes mellitus [20,21,22,23].

Some data obtained in HCV patients treated with DAAs have reported an unexpected increase in the incidence of early HCC in these patients [24,25], while other studies have not systematically confirmed these data. The controversial data have sparked heated debate about the risk of developing HCC after HCV clearance by DAAs and the potential involvement of DAA in liver carcinogenesis [26,27,28].

The purpose of this review is to analyze the published data on the incidence or recurrence of HCC and on the associated risk factors in patients with chronic HCV infection treated with DAAs and SVR, in order to gather an updated overview of the available scientific evidence on such a debated topic. We selected original published studies by searching PubMed, Embase and the Cochrane Library database (through December 2019) using the following keywords: hepatocellular carcinoma, chronic hepatitis C, direct-acting antivirals, occurrence and recurrence. We evaluated the full-text and the references from relevant articles. Moreover, we included the relevant conference proceeding from the 2018 International Congress (AASLD and EASL).

## 2. Occurrence of HCC after Treatment with DAAs

### 2.1. Prospective Studies on the Occurrence of HCC after SVR by DAAs

Table 1 shows the main prospective published studies on the occurrence of HCC after treatment with DAAs. In 2016, our group was the first to report an unexpected occurrence rate of HCC [29] in a cohort of 280 HCV patients after SVR by DAAs. Unfortunately, the few reported cases did not allow a rigorous analysis of the potential risk factors. In two separate research letters, Cardoso et al. [24] and Kobzial et al. [30], two authors from Portugal and Austria, respectively, reported a significant and unexpected high incidence of HCC after clearance of HCV by DAAs. In a series of 54 HCV patients treated with sofosbuvir and ledipasvir for 24 weeks, Cardoso et al. [24] reported that 7.4% of patients were diagnosed with HCC after a median follow-up of 12.0 months (IQR 9.4–12.5 months) from viral clearance. No causative role in HCC development for any considered baseline factor was identified, nevertheless the authors speculated that immune system dysregulation may play a role. Kozbial et al. [30] reported an overall cumulative incidence of de novo HCC after SVR by DAAs in 5.2% of patients during a 48-week follow-up.

In contrast to previous studies, Cheung et al. [31] observed a HCC occurrence rate after SVR by DAAs similar to that observed in untreated HCV patients. Similarly, Foster et al. [32] did not observe significant differences in HCC occurrence rates in the 6 months post SVR in a prospective study including 467 patients treated with DAAs compared with a similar group of untreated HCV cirrhotic patients. In a large Italian cohort of more than 3000 patients undergoing treatment with DAAs, during a median follow-up of 17 months [33], the rate of occurrence of HCC in the sub-cohort of cirrhotic patients was similar to that expected for untreated patients. There was a progressive reduction in the incidence of HCC after the first year in the cirrhotic patients. The authors hypothesized that the progressive decline over time of HCC could be related to the reduction in post-treatment intrahepatic inflammation. Mettke et al. [34] evaluated the incidence of HCC in a cohort of HCV cirrhotic patients after SVR by DAAs as compared with historical data of untreated patients, concluding that treatment with DAAs does not change the short-term risk of HCC. A multivariate analysis revealed that elevated MELD (Model of End-stage Liver Diseases) and alpha-fetoprotein values were independent factors associated with HCC development.

Calvaruso et al. [35] studied a cohort of 2249 consecutive HCV cirrhosis patients treated with DAAs. Seventy-eight patients (3.5%) developed HCC during a mean follow-up of 14 months. In this cohort, low levels of albumin (<3.5 g/dL), a low platelet count (<120 × 10^9^/L) and the absence of SVR were associated with an increased risk of HCC. Recently, a cohort of 9895 HCV patients was evaluated in a French multicenter study by Carrat et al. [36]. The occurrence of HCC in patients treated (*n* = 7344) with DAAs and untreated (*n* = 2551) was 2.5% and 2.8%, respectively, during a median follow-up period of 33.4 months. The authors concluded that treatment with DAAs was associated with a reduced risk of HCC in patients with chronic HCV infection. Similarly, Pinero et al. [37], in a cohort of 784 HCV cirrhotic patients who underwent treatment with DAAs, observed that the cumulative incidence of HCC was 0.03 (CI 0.02–0.05) and 0.06 (CI 0.04–0.08) at 12 and 24 months of follow-up, respectively. SVR was associated with a 73% reduction in the overall relative risk for HCC recurrence. An Egyptian cohort of 2372 patients infected by genotype 4 HCV with advanced liver fibrosis and cirrhosis was prospectively followed for at least 12 months [38]. The overall HCC incidence was 2.3 per 100 patient-years. In patients with cirrhosis, the incidence of HCC was 2.9 per 100 patient-years, while in patients with advanced liver fibrosis, the incidence of HCC was 0.66 per 100 patient-years. Overall, the results showed a reduced incidence of HCC in both patients with cirrhosis or advanced liver fibrosis. 

Recently, Tani et al. [39], who followed for 24 months 1084 HCV patients who achieved an SVR, showed that the incidence of HCC was 0.61%, 1.88%, 2.82% and 3.71% at 6, 12, 18 and 24 months after treatment with DAAs, respectively. Furthermore, they identified age and alfa-fetoprotein levels as the independent predictors of HCC occurrence.

### 2.2. Retrospective Studies on the Occurrence of HCC after Treatment with DAAs

Table 2 shows the retrospective studies on the occurrence of HCC after treatment with DAAs. The first retrospective observation was reported by an Italian study group [25] that analyzed a cohort of cirrhotic patients treated with DAAs. The rate of occurrence of HCC during the first 6 months after treatment was 3.1%, which was higher than that previously observed throughout the natural history of patients with untreated HCV cirrhosis [40]. Likewise, Nakao et al. [41] observed cumulative HCC incidences of 1.7% and 7% at 1 and 2 years after treatment with DAAs, respectively. In a large retrospective cohort study conducted in 22,500 patients with chronic hepatitis C infection treated with DAAs, Kanwal et al. [42] showed a significant 76% reduction in the HCC risk in cirrhotic patients with an SVR compared with non-SVR patients. Cirrhotic patients had a higher annual incidence of HCC after SVR than that observed in non-cirrhotic patients (1.82 vs. 0.34 per 100 person-years; adjusted hazard ratio, 4.73. 95% CI, 3.34–6.68). In a recent multicenter retrospective study, Marino et al. [43] reported an incidence rate of HCC of 3.7% during a median follow-up of 19.6 months post-treatment with DAAs. Basal liver function, the presence of uncharacterized liver nodules, alcohol intake and hepatic decompensation were associated with a higher risk of developing HCC.

Mecci et al. [44] focused on decompensated cirrhotic patients and compared 80 patients with HCC with 165 patients without HCC treated with DAAs and followed for a mean of 32.4 months. The authors concluded that the presence of baseline nonmalignant liver lesions, diabetes and thrombocytopenia increases the risk of HCC, and HCC is associated with a decreased SVR rate.

Recently, Ioannou et al. [45] retrospectively evaluated the incidence of HCC in 29,033 HCV patients treated with DAAs and followed between 2015 and 2019. The results showed that patients with cirrhosis continued to present a high risk of HCC (>2%/year), regardless of whether the FIB-4 score decreases over time, and therefore the authors recommend a long-time surveillance. Meanwhile, among patients without cirrhosis, those with FIB-4 scores ≥3.25 continue to present a relatively high risk of developing HCC, and are therefore deserving of surveillance.

In summary, considering prospective and retrospective studies, the data show that clearance of HCV by DAAs does not have a significant impact on the occurrence of short–medium term HCC, but reduces the risk of medium–long term HCC.

## 3. Comparative Studies between the Therapeutic Regimens Based on DAAs and IFN on the Occurrence of HCC

Several studies compared cohorts of patients treated with DAAs compared with IFN treatment (Table 3). Nagata et al. [46] retrospectively evaluated two cohorts of HCV patients treated with DAAs or IFN, using a propensity score analysis to reduce the bias of the different follow-up after achieving SVR (6.8 vs. 1.8 years). The results of this study showed that the cumulative occurrence rate of 3-year HCC was similar in the two groups (3.3% vs. 1.4%). The cumulative incidence of HCC was significantly lower for patients who achieved SVR in both groups. Similar results have also been reported in a cohort described by Affronti et al. [47], characterized by a high proportion of decompensated cirrhosis. Using a propensity score matching analysis, Nagaoki et al. [48] evaluated the cumulative incidence of HCC in 154 HCV patients with chronic hepatitis or cirrhosis treated with daclatasvir/asunaprevir compared with a historical cohort of 244 patients treated with IFN-based regimens. The data showed that in the two groups treated with DAAs and with IFN, the incidence of HCC at 1, 3 and 5 years of follow-up was 0.6%, 9% and 9% and 0.4%, 3% and 5% (*p* = 0.053), respectively. In a retrospective multicenter analysis involving 15 centers in Belgium, Bielen et al. [49] evaluated 567 HCV patients treated with an IFN-based regimen, 77 treated with PEG-IFN + DAAs between 2008 and 2013 and 490 who received DAAs between 2013 and 2015. The HCC occurrence rate was 1.7% and 1.1% in patients treated with DAAs with and without PEG-IFN, respectively (*p* = 0.540), so no significant difference in early post-treatment onset of HCC in patients treated with DAAs or IFN. Ioannou et al. [50] retrospectively evaluated 62,354 HCV patients who were started on antiviral treatments (DAAs, DAAs plus IFN and IFN). A total of 3271 incident HCC cases were diagnosed during a mean follow-up of 6.1 years. Regardless of the treatment used (IFN, DAAs or the combination of both), SVR was associated with a significant reduction in HCC in cirrhotic patients compared with those who did not achieve SVR. Treatment with DAAs or DAAs plus IFN was not associated with a different HCC risk compared with treatment with IFN. SVR by DAAs was associated with a 71% reduction in HCC risk. Innes et al. [51] evaluated a total of 857 patients, of whom 31.7% received an IFN-free regimen. In a univariate analysis, IFN-free therapy was associated with a significantly increased risk of HCC. However, after the multivariate adjustment for baseline factors (age, ethnicity, Child–Turcotte–Pugh; platelet count; genotype), no significant risk attributable to IFN-free therapy was confirmed.

Using the Electronically Retrieved Cohort of HCV-Infected Veterans database, Li et al. [52] evaluated a large group of 17,836 HCV patients treated with IFN or with DAAs. Among patients with cirrhosis who achieved SVR, neither the incidence rate of HCC-free survival nor HCC was significantly different between the DAAs and IFN groups (21.2 vs. 22.8 per 1000 person-years; *p* = 0.78 and log-rank *p* = 0.17, respectively). Furthermore, the results showed a significantly higher HCC incidence rate in patients with untreated cirrhosis (45.3 per 1000 person-years). In a retrospective study evaluating more than 30,000 patients undergoing treatment with DAAs, Singer et al. [53] showed a significantly lower risk of HCC compared with untreated patients (Odds ratio (HR) = 0.84, 95% CI: 0.73–0.96), or to IFN-treated patients after adjustment by gender, age and stage of the disease.

Nahon et al. [54] collected data from 35 centers in France on 1270 patients with HCV divided into subgroups: (1) treated with DAA (*n* = 336), (2) those who obtained an SVR following an IFN-based regimen (*n* = 495) and (3) those never treated or not compliant with the IFN (*n* = 439). Patients were followed up with ultrasound every six months to detect the onset of HCC. The three-year cumulative incidence of HCC was in groups 1, 2 and 3 of 5.9%, 3.1% and 12.7%, respectively. Compared with patients in the SVR-IFN group, patients in the DAA group were older, had higher diabetes, portal hypertension and impaired liver function. The authors hypothesized that the high occurrence rate of HCC could be related to patient characteristics (age, diabetes, reduced liver function) and lower screening intensity. In a recent meta-analysis that includes 26 patient cohorts, Waziry et al. [55] did not identify a high risk of developing HCC after HCV treatment with DAAs in patients with cirrhosis, but an individual risk reduced by 63%.

In a recent study, Janjua et al. [56] evaluated a large Canadian cohort treated with DAAs compared with a retrospective cohort treated with IFN. Among patients who responded to DAAs treatment, the incidence rate of HCC was 6.9 per 1000 person-years. Among individuals successfully treated with interferon, the incidence rate of HCC was 1.8. The authors concluded that similar to the interferon era, DAA-related SVR is associated with a 70% reduction in HCC risk. 

Teng et al. [57] compared the preventive tertiary effect between DAAs and peg-IFN-RBV in 301 patients with HCV-HCC by a propensity score corresponding to age, tumor staging, HCC treatment modality and cirrhotic status. The results showed that the tertiary prevention effect lasted in the Peg-IFN/RBV arm (*p* < 0.001), but decreased in the DAA arm (*p* = 0.135) compared with untreated patients.

## 4. DAAs Treatment and Recurrence of HCC in HCV Patients

Recently, some concerns have been raised regarding the possible increased risk of recurrence of HCC after treatment with DAAs (Table 4). In a multicenter retrospective study, Reig et al. [58] showed that 27.5% of the 58 patients treated with DAAs had a recurrence of HCC after a median follow-up of 5.7 months after treatment. In the same way, Conti et al. [25] observed that 29% of 59 patients had a recurrence of HCC during the six months of follow-up after treatment with DAAs. Kozbial et al. [30] and Yang et al. [59], in retrospective studies, confirmed a high relapse rate of HCC after treatment with DAAs. A hypothesis to explain the possible high recurrence rates of HCC observed in these cohorts of patients treated with DAAs is a dysregulation of the antitumor immune response after the sudden clearance of HCV that would promote tumor recurrence [60]. On the other hand, in a study from France [61] including HCV patients previously treated for HCC among whom 13 cirrhotic patients received treatment with DAAs and 66 received no treatment, 7.7% of patients treated with DAAs showed a recurrence of HCC, while in the untreated group, 47% showed a relapse of HCC. Therefore, the results of this study did not confirm that patients treated with DAAs had a high recurrence of HCC. Similar results have been reported in a study conducted in Italy [62], which included 31 consecutive HCV cirrhotic patients with HCC after being cured by locoregional or resection treatment and who received DAAs. The median time between treatment with HCC and the start of treatment with DAAs was 19.3 months and the median follow-up period after treatment with DAAs was eight months. The recurrence of HCC was 3.2% and the authors concluded that treatment with DAAs was not associated with an increased risk of recurrent HCC.

In the multicenter North American cohort study [63], 793 patients with HCV-associated HCC were evaluated, of whom 38.3% received DAAs therapy and 61.7% were untreated. HCC recurred in 42.1% of treated and in 58.9% of untreated patients. The authors concluded that DAAs therapy was not associated with an increased overall or early HCC recurrence.

A French study [64] evaluated 68 consecutive HCV patients with apparently cured HCC, of which 34% were treated with DAAs. The recurrence rate among treated and untreated patients was 1.7/100 and 4.2/100 person-months, respectively (*p* = 0.008). The conclusion was that the HCC recurrence rate was significantly lower in patients treated with DAAs than in untreated patients. Similar results were achieved by Imai et al. [65], who identified the SVR by DAA as an independent factor for the prevention of HCC recurrence. Nakano et al. [66] evaluated 459 patients who had HCC for the recurrence rate and to identify the predictors of HCC recurrence after DAAs treatment. In a median time of 29.2 months, 47.2% of patients developed HCC recurrence. The factors associated were the AFP levels and the number of HCC occurrence before the DAAs treatment.

These conflicting results have generated a heated debate. Studies showing high HCC recurrence rates in patients treated with DAAs may suffer from a selection bias due to a failure to detect HCC in patients with impaired liver function who are eligible for anti-HCV treatment because of the safety of DAAs [60,67].

## 5. Role of DAAs and Genesis of the Occurrence or Recurrence of HCC after HCV Clearance

The development of HCC in HCV patients who obtained SVR by DAAs treatment has sparked a wide debate. The high percentage of SVR achieved with DAAs was expected to greatly reduce the possibility of complications, including decompensation and the development of HCC. In many studies, an early appearance of HCC was detected as early as six–nine months after discontinuation of DAAs therapy [29,30,31,33,34,35,36,37]. To explain this event, two prevailing hypotheses have been made: the presence of small HCC nodules already before the start of treatment or the induction of carcinogenesis by the DAAs mediated by largely unknown immunological mechanisms [68,69,70]. In the first case, small nodules may have escaped the ultrasound screening for pre-treatment with DAAs, whereas contrast enhancement examinations (MRI or CT) are reserved for cases in which basic nodules have already been identified [71]. Notably, HCV clearance does not lead to the reversal of an advanced cirrhotic state, which by itself is a potential risk factor for the development of HCC [20,72]. At the same time, the main theories on HCV-related carcinogenesis have highlighted the fundamental role of chronic inflammation induced by the virus [6,7,8,9,10,11,12,13,19]. Contradictions on the occurrence of HCC have emerged in the numerous studies listed above. The first studies conducted on case series showed an unexpected high rate of HCC after SVR. Some authors reported a higher annual risk incidence in the cirrhotic patients (2–8%), speculating on a possible direct role of DAAs [24,25,29,31].

Subsequent studies conducted on a larger series of cases showed a non-increase in the incidence rate or slightly lower occurrence of HCV compared with untreated patients [33,34,35,36]. Other authors have instead tried to compare the occurrence of HCC after treatment with DAAs and treatment with IFN [44,45,46,54]. The interpretation of these latter data appears to be very complex as the two populations present important differences. The IFN-based therapies were essentially performed in patients with chronic hepatitis with cytolytic activity and with mild to severe fibrosis, and the patients were relatively young. Treatment of cirrhotic patients and patients over the age of 65 was a rarer event due to the side effects. A diametrically opposed population is that treated with DAAs. In fact, from 2015 to 2018, mainly cirrhotic patients with an advanced median age were treated with DAAs [73]. In comparative studies in which the differences in the studied populations were adequately controlled and weighed, the risk of HCC was globally similar or lower in patients treated with DAAs compared with those treated with IFN [48,74,75]. Factors related to the stage of liver disease (e.g., Child B, portal hypertension, low platelet count) or comorbidity (e.g., diabetes, alcohol, smoking, age) were the variables most frequently associated with the development of post-treatment HCC [76,77,78,79,80,81,82].

Several hypotheses have been made on the genesis of HCC after treatment with DAAs. The main one is based on the reduction of immunosurveillance in response to the rapid decrease in viral load. This event would lead to a downregulation of the gene and IFN receptors and would allow the development of tumor cell clones. In particular, the interferon-β, interferon-induced protein 44 (IFI44) and the chemokine ligand 10 of the CXC motif (CXCL10) decreased significantly and normalized rapidly at the end of therapy in peripheral blood mononuclear cells [83]. Hengst et al. [84], evaluating the expression level of 22 chemokine and cytokines (IFN 1 and CXCL10), confirmed that HCV infection is associated with interferon system activation by an intrahepatic interferon-stimulated gene expression and showed that HCV clearance by DAAs induces a rapid significant decrease in the majority of altered inflammatory cytokines. Furthermore, Meissner et al. [85] showed a rapid downregulation of IFN-stimulated genes in liver and blood after virus clearance. The IFN is known to have an immunomodulatory and antiproliferative action [86]. Chu et al. [87] studied the possible role of the reduction in the number of natural killer (NK) liver cells and their cytotoxic activity as a possible factor promoting the appearance of rapid HCC. Similarly, Spaan et al. [88] studied the immune effects of viral clearance with DAAs, demonstrating that the treatment reduced the serum levels of NK cell-stimulating cytokines. Villani et al. [89] carried out an interesting study on the levels of vascular endothelial growth factor (VEFGF), interleukin-10 and tumor necrosis factor-alpha during treatment. DAAs administration induced an early increase in serum VEGF and a change in the inflammatory pattern, altering the balance between inflammatory and anti-inflammatory processes and modifying the antitumor surveillance of the host. The modifications return to normal after the end of treatment. Faillaci et al. [90] confirmed these data, demonstrating that the DAEGs-mediated increase in VEGF favored the recurrence/occurrence of HCC in susceptible patients. In fact, they found this correlation in patients who already have an abnormal activation of neo-angiogenic pathways in liver tissues, as demonstrated by an increase in angiopoietin-2.

Debes et al. [91] performed an interesting study and showed a possible role of cytokines. In patients with HCC de novo after treatment with DAAs, a higher value of nine serum inflammatory cytokines (MIG, IL22, TRAIL, APRIL, VEGF, IL3, TWEAK, SCF, IL21) was observed before treatment, assuming their possible role in carcinogenesis [91].

The hypothesis is that anticancer functions of the immune system maintain a weak balance that could be disturbed by the rapid HCV clearance by DAAs. The resulting “shut-down” should have a boosting effect on cancer development due to an abrupt fall of the virus-stimulated immune surveillance. 

## 6. DAAs Treatment and HCC Fate under the Light Spot of Epigenetic Memory

Epigenetic regulations refer to the complex and multilayered mechanisms involved in the variation of a gene expression without a change in the DNA sequence itself [92]. Epigenetics represents a pivotal mechanism of cellular and organ homeostasis, which includes various regulatory events, such as DNA methylation, histone modifications and noncoding RNAs [93,94]. Changes are an integral part of the epigenetic code and are necessary for the regular development of tissue gene expression in different types of mammalian cells [95]. 

The involvement of epigenetic alterations in the initiation, progression and fate of carcinogenesis associated with chronic HCV infection is well known [94,95]. Despite that HCV virus does not replicate in the cellular nucleus, it has been demonstrated how dramatic the epigenetic impact of HCV is on histone tail modification and DNA methylation [95]. 

Recently, Hamdane et al. [95] performed a genome-wide ChIP mentation-based ChIP-Seq and RNA-seq analyses to characterize histone H3K27ac in liver tissues of patients with chronic HCV infection, HCV patients treated with DAAs or IFN and uninfected subjects. The results of this study showed changes in H3K27ac in the liver tissue of chronic HCV patients and that these changes were still present after HCV clearance by either DAAs or IFN, suggesting that epigenetic alterations caused by HCV persisted after the virus elimination [96].

An analysis of both patients’ liver tissue and mice humanized livers showed that HCV-related epigenetic changes were associated with an increased risk of HCC development [95]. Data were validated in a cohort of HCV cirrhotic patients and in a group whose virus had been eliminated by DAA treatment [95].

Data showed that both the direct interaction between HCV and hepatocytes and indirect mechanisms, such as liver fibrosis development, were involved in the epigenetic changes observed in these patients and, therefore, in patients with advanced fibrosis, the increase in HCC risk persisted even after the virus elimination by DAA [95]. It has also been suggested that the different mechanisms of epigenetic changes may explain why some patients develop HCC even in the absence of significant fibrosis [95].

Epigenetic modifications have also been found in the liver tissue next to the HCC nodule, which seems to suggest that epigenetic modifications precede hepatocarcinogenesis.

The authors argue that there is strong evidence that HCV-induced H3K27ac alterations are causative factors for the risk of HCC development after HCV clearance by DAAs. Among these data, we may find the altered expression of known genes promoting carcinogenesis, the correlation between epigenetic alterations with the Cox score for the HCC risk, the correlation between epigenetic alterations and stage of fibrosis, which is a risk factor for HCC, and H3K27ac alterations in HCC tumors of the same patients [95].

The findings of the study, as pointed out by the authors [95], have a double potential implication: on the one hand, there is the opportunity to develop early plasma biomarkers to identify patients at risk of HCC, i.e., tests that can detect the epigenetic changes of circulating DNA linked to histone complexes; on the other hand, to identify therapeutic targets towards which specific drugs can be developed and be potentially useful in preventing HCC.

Perez et al. [96] showed that HCV infection affects the histone tails post translational modification. Indeed, they observed a strong variation after HCV infection in Huh7.5 cells of Histone 3 Lysine 4 tri-methylation (H3K4Me3), Histone 3 Lysine 9 acetylation (H3K9Ac) and Histone 3 Lysine 9 tri-methylation (H3K9Me3). Those histone modifications are representative of open chromatin: H3K4Me3 and H3K9Ac; and close chromatin H3K9Me3. Those alterations were over more than 1200 genomic regions for H3K4Me3, more than 3800 genomic regions H3K9Ac and over 9000 genomic regions for H3K9Me3. On the whole, these results underlined the impact of HCV infection from an epigenetic point of view. HCV infection induces epigenetic alterations by modifying their gene expression and affecting the molecular expression of the virus life cycle and HCC development [96]. Noteworthy is the observation that anti-HCV treatment with DAAs is able to clear the virus from the host, but not to restore the concomitant epigenetic signatures associated with the risk of HCC [96]. The data suggest that when the infection has induced epigenetic changes, the gene expression is preserved in the cells, and therefore the presence of the virus is no longer necessary to exert oncogenic effects on the host cells. In a small group of patients recovered from HCV, epigenetic changes have been shown to persist and associate with the evolution of HCC. Data from this study also show that, unlike for DAAs, subjects who recovered with the use of interferon do not show the persistence of epigenetic modifications induced by HCV. On this basis, the authors suggest this as the potential reason for explaining the observation that HCC development is more frequent after DAAs treatment than interferon-based treatment [97].

Significant is the demonstration that in vitro treatments with drugs such as C646, a specific inhibitor of H3K9Ac, or Erlotinib, a specific inhibitor of the epidermal growth factor receptor (EGFR) signaling pathway involved in cancer invasion and metastasis, as well as in the regulation of gene expression, restored the epigenetic alterations, thus preventing oncogenesis. The authors postulated that treatments restoring epigenetic changes may have potential implications in preventing the development of HCC (96).

Several landscapes of human genome were found to be methylated by HCV infection. Deng et al. [98] reported the alteration in gene regulatory elements via methylation status in HCC derived from HCV infected patients, whilst Wijetunga et al. [99] reported that HCV infection determines a hyper-methylation of regulatory enhancers and the polycomb target gene. This methylation of the genome correlated with specific genes’ downregulation in both a direct and indirect manner. On the other hand, epigenetic perturbation due to HCV infection is mainly represented by histone tails PTM. Despite the virus eradication, the gene pattern with the epigenetic alteration is preserved, with a direct consequence in the development of HCC. 

The variations in the histone and of the methylation landscape should define a new profile and personalized medicine for HCV patients, particularly after SVR by DAAs treatment. However, further studies are needed to identify biomarkers for post-SVR HCC, as well as to develop epigenetic target therapies based on a more complete understanding of the molecules and pathways involved in hepatocarcinogenesis. 

## 7. Predictive Factors of HCC Appearance or Recurrence

In Figure 1, the predictors of HCC development after HCV clearance by DAAs are illustrated. 

Conti et al. [25] assessed risk factors for HCC in 344 cirrhotic patients after HCV clearance by DAAs. The analysis showed that the presence of advanced cirrhosis and the history of previous HCC were independent factors associated with the development of HCC. Similarly, in a large multicenter study that included 1675 HCV patients followed for 17 months post-treatment with DAAs, it was shown that the 1-year cumulative rates of HCC recurrence were 6.5% and 23.1% for the non-cirrhosis and cirrhosis patients, respectively, and cirrhosis was identified as a risk predictor of HCC [100]. Basal liver stiffness could be considered as a potential predictive factor for a HCC appearance in this setting. To test this hypothesis, we conducted a prospective study [101] on 258 HCV cirrhotic patients treated with DAAs, measuring basal liver stiffness (LS) by Fibroscan^®^. Patients were divided into three groups, based on their liver stiffness: <20 kPa (*n* = 72), between 20 and 30 kPa (*n* = 92) and >30 kPa (*n* = 94). The results showed a statistically significant increased HCC risk in the LS > 30 kPa group (*p* = 0.019; HR 0.329; 95% CI 0.131–0.830). The ROC curve analysis identified a cutoff value of LS of 27.8 kPa, with the highest sensitivity and specificity (72% and 65%, respectively), to individuate patients at high risk of HCC development. 

Degasperi et al. [102] showed similar results in a retrospective study on 505 cirrhotic patients DAAs treated, demonstrating that a baseline LS > 30 kPa was an independent predictor of de novo HCC, being that the 3-year estimated incidence of de novo HCC was 20% in patients with LSM > 30 kPa versus 5% in patients with LSM ≤ 30 kPa (*p* = 0.0003). Recently, in a population of 640 HIV/HCV co-infected patients followed for a median period of 31.6 months post-SVR by DAAs, with the aim of verifying the incidence of HCC, it was shown that none of the 374 patients with LS < 14 kPa at baseline developed HCC [103]. The authors suggest that the LS measurement may be helpful in selecting patients who may not be subjected to a tight surveillance program.

In a large study that included 1022 consecutive patients treated with DAAs, the risk predictors of HCC were assessed during a 48-month follow-up period. The analysis of the data showed that an early HCC occurrence was independently associated with cirrhosis, the male gender and diabetes [104]. Furthermore, this paper also showed that sofosbuvir-based therapy without ribavirin was associated with an HCC occurrence 5.7 times higher than those with other treatment schedules and was an independent factor of HCC development. The authors point out that cirrhotic patients receiving sofosbuvir without ribavirin should undergo a careful and early follow-up [104].

Several studies have shown that the male sex, diabetes and elevated serum alpha-fetoprotein levels before and after DAAs treatment are significant predictors of the onset or recurrence of HCC [34,39,66,67,102].

## 8. Conclusions

HCC still represents a serious complication of cirrhosis with a significant mortality rate [105]. In most of the studies reviewed, SVR by DAAs in HCV patients does not appear to have an impact on the occurrence and recurrence rate of HCC in the short-term post-viral clearance, suggesting that careful surveillance of HCC in patients with cirrhosis should be mandatory. On the other hand, studies conducted on large numbers of HCV patients treated with DAAs showed a reduction in risk of development of HCC in the medium to long term. Comparative studies between DAAs regimen and IFN-based therapies did not provide solid data for the many differences in the population compared, however, treatment with DAAs does not appear to be less effective than IFN-based regimens in reducing the incidence of HCC. 

The underlying mechanisms of HCC occurrence despite a viral clearance are presently unknown, with no clear evidence supporting a tumorigenic role of DAAs, which remains a controversial hypothesis that must be necessarily verified by ad hoc studies. Moreover, the epigenetic modulation of HCV virus in host cells should be considered, according to emerging data of recent studies. 

Follow-up strategies must reflect these uncertainties, so careful ultrasound monitoring of cirrhotic patients is a convincing measure to be taken after HCV clearance by DAAs. The surveillance must be carried out with particular attention and at close times in the post-treatment of subjects presenting the risk factors highlighted so far. Future studies on epigenetics could identify useful therapeutic targets in the prevention of HCC.

## Figures and Tables

**Figure 1 cancers-12-01351-f001:**
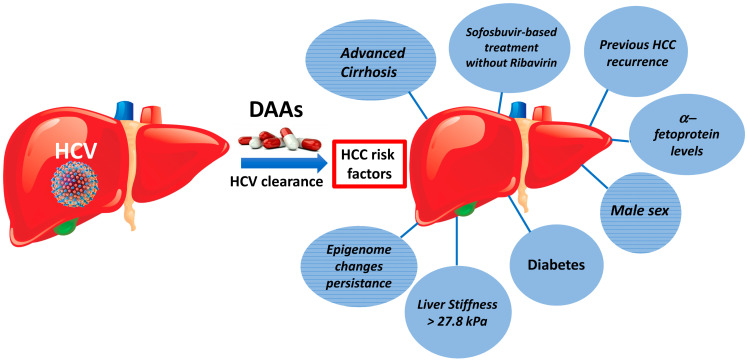
Risk factors of occurrence or recurrence of HCC after HCV clearance by DAAs treatment.

**Table 1 cancers-12-01351-t001:** Prospective study of hepatocellular carcinoma (HCC) occurrence after rapid virologic response (SVR) by direct-acting antivirals (DAAs).

Authors	Year	Country	Study Design	Sample Size	Control Group	Follow-Up (Months, Median)	HCC Occurrence Rates (%)
Rinaldi [29]	2016	Italy	Prospective; cirrhosis	280	NA	3	3.2
Cheung [31]	2016	United Kingdom	Prospective; Decompensated cirrhosis	406	Untreated	18	5.4
Foster [32]	2016	United Kingdom	Prospective; cirrhosis and decompensated cirrhosis	467	Untreated	6	5.4
Romano [33]	2018	Italy	Prospective; cirrhosis	3917	NA	7.4	2.1 (year)
Mettke [34]	2017	Germany	Prospective; cirrhosis	158	Untreated	15	3.7
Calvaruso [35]	2018	Italy	Prospective; cirrhosis	2249	NA	14	3.5
Carrat [36]	2019	France	Prospective; Cirrhosis and non-cirrhosis	7344	Untreated	33.4	4.3
Pinero [37]	2019	Latin America	Prospective; Cirrhosis and non-cirrhosis	784	NA	16	3 (year)
Shina [38]	2020	Egypt	Prospective; cirrhosis	2372	NA	23.6	2.34 (year)
Tani [39]	2020	Japan	Prospective; Cirrhosis and non-cirrhosis	1084	NA	24	3.71

NA: not available.

**Table 2 cancers-12-01351-t002:** Retrospective study of HCC occurrence after SVR by DAAs.

Authors	Year	Country	Study Design	Sample Size	Control Group	Follow-Up (Months, Median)	HCC Occurrence Rates (%)
Cardoso [24]	2016	Portugal	Research letter; Cirrhosis	54	NA	12	7.4
Conti [25]	2016	Italy	Retrospective; Cirrhosis	344	NA	6	3.1
Kozbial [30]	2016	Austria	Research letter; cirrhosis	195	NA	12	6.6
Nakao [41]	2017	Japan	Retrospective; Cirrhosis and non-cirrhosis	242	NA	15	2.8
Kanwal [42]	2017	USA	Retrospective; Cirrhosis and non-cirrhosis	22,500	NA	*	*
Marino [43]	2019	Spain	Retrospective; Cirrhosis and non-cirrhosis	1123	NA	19.6	3.7
Mecci [44]	2019	United Kingdom	Retrospective Cirrhosis	245	NA	32.4	*
Ioannou [45]	2019	USA	Retrospective; cirrhosis	48,135	NA	64.8	3.66

* SVR vs. non-SVR: 0.90 vs. 3.45 HCC per 100 person-years; aHR, 0.28, 95% CI = 0.22–0.36). NA: not available.

**Table 3 cancers-12-01351-t003:** HCC occurrence compared to DAAs and IFN-based regimens.

Authors	Year	Country	Study Design	Sample Size (DAAs/IFN)	DAAs: HCC Occurrence Rates (%)	IFN: HCC Occurrence Rates (%)	*p*
Nagata [46]	2017	Japan	Retrospective	752/1145	3.3	1.4	0.49
Nagaoki [48]	2017	Japan	Retrospective	154/244	0.6 (1st year)	0.4 (1st year)	0.053
Bielen [49]	2017	Belgium	Retrospective	490/77	1.7	1.1	0.54
Ioannou [50]	2017	USA	Retrospective	21,948/35,871	No difference		
Innes [51]	2018	United Kingdom	Retrospective	272/585	No difference after multivariate adjustment		0.744
Li [52]	2018	USA	Retrospective	5834/3534	No difference		
Singer [53]	2018	USA	Retrospective	30,183/12,948	DAAs treatment was associated with a reduced risk		
Nahon [54]	2018	France	Retrospective	336/495	5.9	3.1	0.001
Waziry [55]	2017	Australia	Meta-analyses	13,875	No difference		
Janjua [56]	2020	Canada	Retrospective	3905/8871	6.9/1000 PY	1.8/1000 PY	
Teng [57]	2019	Taiwan	Retrospective	79/102	0.38	0.56	0.186

**Table 4 cancers-12-01351-t004:** Prospective and retrospective study of HCC recurrence after SVR by DAAs.

Authors	Year	Country	Study Design	Sample Size	Control Group	Follow-Up (Months, Median)	HCC Recurrence Rates (%)
Reig [58]	2016	Spain	Retrospective	58	NA	5.7	27.6
Conti [25]	2016	Italy	Retrospective	59	NA	6	28.8
Kozbial [30]	2016	Austria	Research letter	22	NA	7	86
Yang [59]	2016	USA	Prospective	18	NA	NA	27.8
Pol [61]	2016	France	Prospective	13	66	16.5	7.7
Zavaglia [62]	2017	Italy	Research letter	31	NA	8	3.2
Singal [63]	2019	USA	Retrospective	304	489	10.4	42.1
Virlogeux [64]	2017	France	Retrospective	23	45	13	47.8
Imai [65]	2020	Japan	Retrospective	13	64	36	23.8
Nakano [66]	2019	Japan	Prospective	459	NA	29.4	47.2

NA: not available.

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
