# Peer review of "Risk of Hepatocellular Carcinoma after HCV Clearance by Direct-Acting Antivirals Treatment Predictive Factors and Role of Epigenetics"

_cancers, 2020, doi:10.3390/cancers12061351_

Round 1
Reviewer 1 Report
In the review from Rinaldi L. and coll. entitled “Risk of hepatocellular carcinoma after HCV clearance by direct-acting antivirals treatment: the role of drugs and of epigenetics”, authors summarized the recent findings about HCC occurrence and recurrence after DAA treatment.
Despite the intriguing title, the review marginally and superficially deals with possible epigenetic mechanisms involved in HCC occurrence after DAA treatment (line 277 to line 335), thus the review loses some of the potential interest. In addition I was not able to understand to which “drug” authors are referring when saying “:the role of drugs”. For these reasons the review is not so different from many others recently published in the field.
In addition there are some major revisions
- In the abstract authors mentioned that “SVR has no significant impact on the occurrence of HCC in the short and medium term”, this is an important aspect, not enough emphasized in the text, I suggest to report some data on this regard, taking into consideration the differences among the studied patients (presence/absence of cirrhosis, for example)
- In tables, I suggest to include some relevant clinical factors characterizing patients, such as the presence/absence of liver cirrhosis. This would be important in order to better understand differences in occurrence and recurrence rates among studies. For example, in Kanwal’s study, the reported HCC occurrence rate is 0.9 per year, this includes both patients with and without cirrhosis. If patients with cirrhosis are considered separately, the HCC occurrence rate is different. Thus, this number is not comparable with the ones obtained from studies considering cirrhotic patients only, since cirrhosis is a well-known pre-neoplastic condition.
- Section 4. It worth to be notice that some of the studies about the HCC recurrence rates after DAA therapy lack of important information, such as time to recurrence, type of treatment, response to treatment, the method used to define the response to treatment, and so on. These are some limitations that make hard the comparison among studies. I suggest authors to report those data when present, also as supplementary material, it would be useful and something distinguishing the review from others.
- Section 5. I suggest to change the title since there is no mechanism proposed. It is a summary of deregulated cytokines and chemokines from which some speculations about the involvement of imuno surveillance are made.
- Figure 1 is not showing the mechanism, it is just reporting elements/factors influencing occurrence/recurrence, such as low platelets (it is not a mechanism), Child-Pugh B (is not a mechanism), in addition cytokine dysregulation may be involved in the immune-escape.
- Section 7 it is not useful except for self-citation purposes
- Line 280 -284, data from Hengst study are not well reported, the paragraph is not clear, and includes some grammar mistakes. In addition, which is the meaning of the study by Heighst and coll? How it is linked with the hypothesis of reduced immunosurveillance?
Minor revisions
- Line 23, abstract, there is a double space after “long term.”
- Lines 27-28 “Moreover, it should be considered the epigenetic modulation of HCV virus in the host cells. Some histone tail modification and DNA methylation are kept even after the virus eradication.” Please check the grammar
- Line 39 “Although DAAs are likely to affect the immune response, the underlying biological mechanisms remain DAAs largely unexplored” cryptic please rephrase. In addition, please use a more hypothetical form in “DAAs are likely to affect”.
- Line 40, Ref 13 is not appropriate please change
- Line 44, can authors provide a reference about the relation between HCC and reduced immune surveillance in HCV-infected patients?
- Lines 47,48 “Hence, HCV clearance by DAAs and the consequent reduction of hepatic necro- inflammation activity are expected to prevent HCC development.”. It depends from the severity of the disease, some patients have an already compromised liver, thus I suggest to modify the sentence, instead of “prevent” maybe authors can use “reduce the risk” or something similar.
- Lines 58-60 please check the English grammar
- Line 72, there is a space after “[32]”
- Line 72, “Two authors from Portugal and Austria respectively, reported P, a significant”, not clear
- Line 75, Maybe should be mentioned that the medium time for HCC development was 7.6 months. This would be in line with what reported in the abstract.
- Table 1, please substitute “,” with “.”
- Line 92 “Mettke et al. [36] studied the short-term incidence of HCC in a cohort of HCV cirrhotic patients with a SVR by DAAs comparared to historical data of untreated patients and concluded” please check grammar
- Line 141, a full stop is missing
- Lines 144,145 please check the grammar
- Lines 160 and 167 a double space is present
- Lines 184 and 185 please correct “an SVR” with “a”
- Line 188 a comma should be added between “were older” and “had diabetes”
- Lines 191-193 please could you explain better?
- Line 196, formatting
- Lines 212-215 please simplify the sentence “On the other hand…relapse of HCC”
- Line 279, which is the “gene” the authors are referring to? In addition ref 84 is not appropriate.
- Line 319 some double spaces are present
- Line 330 a double space is present
Author Response
Point by point reply
Reviewer 1
Thank you very much for the revision of the manuscript. We appreciated the criticisms and suggestions that we have accepted in full and in agreement we have reviewed the paper. Thank you, because with your suggestions our paper has certainly improved a lot.
General Comments
- In the review from Rinaldi L. and coll. entitled “Risk of hepatocellular carcinoma after HCV clearance by direct-acting antivirals treatment: the role of drugs and of epigenetics”, authors summarized the recent findings about HCC occurrence and recurrence after DAA treatment.
Despite the intriguing title, the review marginally and superficially deals with possible epigenetic mechanisms involved in HCC occurrence after DAA treatment (line 277 to line 335), thus the review loses some of the potential interest. In addition I was not able to understand to which “drug” authors are referring when saying “:the role of drugs”. For these reasons the review is not so different from many others recently published in the field.
Replay. Thank you for the comments. We agree with the referee that the topic of epigenetics, which should be an important and innovative point of our work, has been treated too tightly. In the review, the epigenetic topic was treated with greater accuracy and all the advantages that may derive from the studies made and the future prospects were indicated in detail (see paragraph 6, revised).
In relation to the title we agree with her that it is not clear and does not express exactly what we wanted to communicate. Therefore, the title has been changed and reflects more appropriately to the content and message.
In addition there are some major revisions
- In the abstract authors mentioned that “SVR has no significant impact on the occurrence of HCC in the short and medium term”, this is an important aspect, not enough emphasized in the text, I suggest to report some data on this regard, taking into consideration the differences among the studied patients (presence/absence of cirrhosis, for example)
- Replay: Thanks, we agree with the referee on the point underlined. In the review of the work, we better underlined the aspect in the text, also in relation to the presence or absence of cirrhosis, when we discussed the prospective retrospective studies (points 2.2 and 2.2, revised). In addition, a conclusion has been added at the end of section 2.2 to emphasize the important aspect.
- In tables, I suggest to include some relevant clinical factors characterizing patients, such as the presence/absence of liver cirrhosis. This would be important in order to better understand differences in occurrence and recurrence rates among studies. For example, in Kanwal’s study, the reported HCC occurrence rate is 0.9 per year, this includes both patients with and without cirrhosis. If patients with cirrhosis are considered separately, the HCC occurrence rate is different. Thus, this number is not comparable with the ones obtained from studies considering cirrhotic patients only, since cirrhosis is a well-known pre-neoplastic condition.
- Thanks. We agree with the referee, although most of the studies have been conducted in cirrhotic patients. However, in agreement with the referee we have included the condition with or without cirrhosis in the various studies in tables 1 and 2. We also reported some differences in the incidence of HCC in the two conditions in the text.
- Section 4. It worth to be notice that some of the studies about the HCC recurrence rates after DAA therapy lack of important information, such as time to recurrence, type of treatment, response to treatment, the method used to define the response to treatment, and so on. These are some limitations that make hard the comparison among studies. I suggest authors to report those data when present, also as supplementary material, it would be useful and something distinguishing the review from others.
Replay: Thanks. We have reported all possible information in the tables and text. However, we agree with the referee that in relation to the different conditions that the various studies have been conducted it is not easy to compare the results. However, beyond the single result, the final message remains, namely the impact that the treatment has on the development of HCC.
- Section 5. I suggest to change the title since there is no mechanism proposed. It is a summary of deregulated cytokines and chemokines from which some speculations about the involvement of imuno surveillance are made.
Replay: Thanks. We have changed the title of the paragraph by replacing "mechanisms" with "genesis" to avoid confusion
- Figure 1 is not showing the mechanism, it is just reporting elements/factors influencing occurrence/recurrence, such as low platelets (it is not a mechanism), Child-Pugh B (is not a mechanism), in addition cytokine dysregulation may be involved in the immune-escape.
Replay: Thanks for the observation. We agree that the figure can be confusing as it is a mixed figure. We also realize it does not add much to the purpose of the work as it refers to the general conditions associated with HCC, therefore we have decided to eliminate the figure. However, we have added a different figure below that embodies the spirit of the work without creating confusion (see later).
- Section 7 it is not useful except for self-citation purposes
Replay: Thanks for the observation. We agree with the referee as the paragraph is written it could seem mainly a self-quote, this was not our intention and we apologize, although we had described the predictive factors through the text. We think that the paragraph is one of the central points of the work, therefore we have rewritten the chapter, discussing the works that have analyzed the predictive efactors. In consideration of the high number of patients and the high costs of follow-up, to facilitate the selection of patients who require particular attention in their management, we have made a figure 1 that reaches all the main information.
- Line 280 -284, data from Hengst study are not well reported, the paragraph is not clear, and includes some grammar mistakes. In addition, which is the meaning of the study by Heighst and coll? How it is linked with the hypothesis of reduced immunosurveillance?
Replay: Thanks for the observation. We have provided the information and corrected the errors. The work shows that HCV infection is associated with an immunosurveillance disorder and that after the clearance of HCV by DAAs, rapid dysregulation of immunity is observed. This is the basis for those who think that the condition may favor the early development of HCC, as better explained in the text.
Minor revisions
- Line 23, abstract, there is a double space after “long term.”
- Replay: Thanks, done
- Lines 27-28 “Moreover, it should be considered the epigenetic modulation of HCV virus in the host cells. Some histone tail modification and DNA methylation are kept even after the virus eradication.” Please check the grammar
- Replay: Thanks, done. Sorry.
- Line 39 “Although DAAs are likely to affect the immune response, the underlying biological mechanisms remain DAAs largely unexplored” cryptic please rephrase. In addition, please use a more hypothetical form in “DAAs are likely to affect”.
- Replay: Thanks, done
- Line 40, Ref 13 is not appropriate please change
- Replay: Thanks, done. Sorry.
- Line 44, can authors provide a reference about the relation between HCC and reduced immune surveillance in HCV-infected patients?
- Replay: Thanks, done. See ref. 16
- Lines 47,48 “Hence, HCV clearance by DAAs and the consequent reduction of hepatic necro- inflammation activity are expected to prevent HCC development.”. It depends from the severity of the disease, some patients have an already compromised liver, thus I suggest to modify the sentence, instead of “prevent” maybe authors can use “reduce the risk” or something similar.
- Replay: Thanks, done
- Lines 58-60 please check the English grammar
- Replay: Thanks, done
- Line 72, there is a space after “[32]”
- Replay: Thanks, done
- Line 72, “Two authors from Portugal and Austria respectively, reported P, a significant”, not clear
- Replay: Thanks, we eliminate "reported P" was a mistake.
- Line 75, Maybe should be mentioned that the medium time for HCC development was 7.6 months. This would be in line with what reported in the abstract.
- Replay: Thanks, done
- Table 1, please substitute “,” with “.”
- Replay: Thanks, done
Line 92 “Mettke et al. [36] studied the short-term incidence of HCC in a cohort of HCV cirrhotic patients with a SVR by DAAs comparared to historical data of untreated patients and concluded” please check grammar
- Replay: Thanks, done
- Line 141, a full stop is missing
- Replay: Thanks, done
- Line 144,145 please check the grammar
- Replay: Thanks, done
- Lines 160 and 167 a double space is present
- Replay: Thanks, done
- Lines 184 and 185 please correct “an SVR” with “a”
- Replay: Thanks, done
- Line 188 a comma should be added between “were older” and “had diabetes”
- Replay: Thanks, done
- Line 191-193 please could you explain better?
- Replay: Thanks, done
- Line 196, formatting
- Replay: Thanks, done
- Lines 212-215 please simplify the sentence “On the other hand…relapse of HCC”
- Replay: Thanks, done
- Line 279, which is the gene the authors are referring to? In addition ref 84 is not appropriate.
- Replay: Thank. Sorry the reference was inappropriate and has been deleted. The variations are specified in the following lines of the text, in particular the interferon signaling pathway.
- Line 319 some double spaces are present
- Replay: Thanks, done
- Line 330 a double space is present
- Replay: Thanks, done

Reviewer 2 Report
Rinaldi et al. submitted a review about "Risk of hepatocellular carcinoma after HCV clearance by direct-acting antivirals treatment: the role of drugs and of epigenetics". Please find below minor comments:
1) The authors could indicate how many articles have been retrieved from PubMed, Embase and Cochrane Library Database.
2) Line 195: INF -> IFN
3) Line 283: of 22 chemokine cytokines ->22 chemokines and cytokines
4) line 325: iper-methylation -> hyper-methylation
5) line 328: Could the author clarify the sentence: In the fascinating paper ... the disruptive evidence that epigenetic deregulations have occurred upon infection.
Author Response
Revisor 2
Thank you very much for the revision of the manuscript.
Comment
The authors could indicate how many articles have been retrieved from PubMed, Embase and Cochrane Library Database.
Replay: Thanks. We have done a research by consulting all the works on the topic of the last 3-4 years. We didn't do a systematic review, so we didn't do a PRISMA check list. We analyzed and commented on all works published in journals with reviewers who were of interest to the purpose of the work. The papers mentioned in the references were those taken in particular attention.
2) Line 195: INF -> IFN
Replay: Thanks. Done
3) Line 283: of 22 chemokine cytokines ->22 chemokines and cytokines
Replay: Thanks. Done
4) line 325: iper-methylation -> hyper-methylation
Replay: Thanks. Done
5) line 328: Could the author clarify the sentence: In the fascinating paper ... the disruptive evidence that epigenetic deregulations have occurred upon infection.
Replay: Thanks. Done. The sentence has been made clearer
Reviewer 3 Report
The review by Rinaldi et al. aims at systematically analyzed the impact of SVR by DAAs on the risk of occurrence or recurrence of HCC. The article flows well and analyzes in succession relevant literature about i) prospective studies on the occurrence of HCC after SVR by DAAs, ii) Retrospective studies on the occurrence of HCC after treatment with DAAs, iii) Comparative studies between the therapeutic regimens based on DAAs and IFN on the occurrence of HCC, iv) DAAs treatment and recurrence of HCC in cirrhotic HCV patients and v) Role of DAAs and proposed mechanisms in the occurrence or recurrence of HCC after HCV clearance. I have important concerns:
A) systematic review must follow strict and precise rules dictated by the PRISMA check list: http://prisma-statement.org/PRISMAStatement/FlowDiagram. Rules of inclusion/exclusion of the discussed studies are not described. The choice of the "main" papers to discuss seem arbitrary. The authors can choose not to do a systematic review. In this case the phrase in the abstract "In this review, we systematically analyzed.." is not appropriate. In any case, I ask the authors to perform a systematic review - and provide the relative chart - as this is topic of great interest for a wide readership.
B) The authors must take out the paragraph about epigenetics, as it is totally disjointed with the rest, on top of being largely not inclusive of the current literature.
Author Response
Revisor 3
Thank you very much for the revision of the manuscript.
- A) systematic review must follow strict and precise rules dictated by the PRISMA check list: http://prisma-statement.org/PRISMAStatement/FlowDiagram. Rules of inclusion/exclusion of the discussed studies are not described. The choice of the "main" papers to discuss seem arbitrary. The authors can choose not to do a systematic review. In this case the phrase in the abstract "In this review, we systematically analyzed.." is not appropriate. In any case, I ask the authors to perform a systematic review - and provide the relative chart - as this is topic of great interest for a wide readership.
Replay: Thanks, we have done a large review of the literature, but in agreement with the referee we cannot say that it was systematic. So in the abstract and in the text it was better specified
- B) The authors must take out the paragraph about epigenetics, as it is totally disjointed with the rest, on top of being largely not inclusive of the current literature.
Replay. Thanks for the comment. In accordance with referee 1, we think that the epigenitic aspect may be of great interest as well as of great future prospect. Furthermore, this may be one of the innovative aspects of the review. Given the complexity of the topic, there are currently not many works in this regard. Although the existing ones are of great impact and show how epigenetic alterations persist after the clearance of HCV by DAAs and are implicated in liver carcinogenesis. They also show the future possibility of development of biomarkers that can identify patients at risk, but also the possibility of developing drugs with epigenetic targets that can be useful in the prevention of HCC. In the review of the paper, in accordance with referee 1, we have expanded the chapter on the role of epigenetics by making it less hermetic and more accessible to general understanding.
Round 2
Reviewer 1 Report
I appreciate the effort authors put in fullfilling al the requests. I think that now the Review have improved.
Reviewer 3 Report
None.